# Equivocating and Deliberating on the Probability of COVID-19 Infection Serving as a Risk Factor for Lung Cancer and Common Molecular Pathways Serving as a Link

**DOI:** 10.3390/pathogens13121070

**Published:** 2024-12-06

**Authors:** Abdelbasset Amara, Saoussen Trabelsi, Abdul Hai, Syeda Huma H. Zaidi, Farah Siddiqui, Sami Alsaeed

**Affiliations:** 1Department of Medical Laboratory Technology, Faculty of Applied Medical Sciences, Northern Border University, Arar 91431, Saudi Arabia; abdulhai.agha@nbu.edu.sa (A.H.); farah.khan@nbu.edu.sa (F.S.); sami.alsaeed@nbu.edu.sa (S.A.); 2Center for Health Research, Northern Border University, Arar 91431, Saudi Arabia; sawsan.trablsi@nbu.edu.sa; 3Department of Community Health, Faculty of Applied Medical Sciences, Northern Border University, Arar 91431, Saudi Arabia; 4Department of Chemistry, Faculty of Science, Northern Border University, Arar 91431, Saudi Arabia; syeda.zaidi@nbu.edu.sa

**Keywords:** SARS-CoV-2, COVID-19 long-term effect, lung cancer, risk factor, pathway

## Abstract

The COVID-19 infection caused by SARS-CoV-2 in late 2019 posed unprecedented global health challenges of massive proportions. The persistent effects of COVID-19 have become a subject of significant concern amongst the medical and scientific community. This article aims to explore the probability of a link between the COVID-19 infection and the risk of lung cancer development. First, this article reports that SARS-CoV-2 induces severe inflammatory response and cellular stress, potentially leading to tumorigenesis through common pathways between SARS-CoV-2 infection and cancer. These pathways include the JAK/STAT3 pathway which is activated after the initiation of cytokine storm following SARS-CoV-2 infection. This pathway is involved in cellular proliferation, differentiation, and immune homeostasis. The JAK/STAT3 pathway is also hyperactivated in lung cancer which serves as a link thereof. It predisposes patients to lung cancer through myriad molecular mechanisms such as DNA damage, genomic instability, and cell cycle dysregulation. Another probable pathway to tumorigenesis is based on the possibility of an oncogenic nature of SARS-CoV-2 through hijacking the p53 protein, leading to cell oxidative stress and interfering with the DNA repair mechanisms. Finally, this article highlights the overexpression of the SLC22A18 gene in lung cancer. This gene can be overexpressed by the ZEB1 transcription factor, which was found to be highly expressed during COVID-19 infection.

## 1. Introduction

Abbreviated as COVID-19, Coronavirus Disease 2019 is a condition caused by the Severe Acute Respiratory Syndrome Coronavirus 2 (SARS-CoV-2) [1]. The global pandemic began in Wuhan, China, where COVID-19 was first identified in December 2019 and swiftly spread worldwide [2]. SARS-CoV-2 is an enveloped RNA virus. It has a single strand RNA and pertains to the coronavirus family. This family of viruses is known to cause zoonotic diseases, for instance Severe Acute Respiratory Syndrome (SARS) and Middle East Respiratory Syndrome (MERS) [3]. The virus primarily targets the respiratory system, leading to a range of infections from asymptomatic to severe respiratory failure and collapse. COVID-19 clinical manifestations vary from mild symptoms (flu-like) to severe pneumonia, Acute Respiratory Distress Syndrome (ARDS), and multi-organ failure, particularly in high-risk populations suffering from comorbidities and the elderly [1].

COVID-19 evolved from being an endemic to becoming a global pandemic, rapidly spreading across the globe and eventually impacting all aspects of life. As of 2024, about 760 million cases and about 7 million deaths have been reported globally [4]. Following guidelines from the Centers for Disease Control (CDC) and WHO, governments implemented and thereby enforced public health measures which included lockdowns, quarantines, and social distancing to curb the transmission of the virus [5]. Mass vaccination campaigns of massive proportions were undertaken after the conclusion of clinical trials and the development of mRNA vaccines, and while these interventions significantly reduced severe disease and death, the pandemic perpetually persists in various parts of the world due to emerging variants of concern such as Delta and Omicron [6]. Beyond the direct toll, the pandemic has had profound socio-economic effects, while exacerbating the strained healthcare systems globally [7].

Initially, SARS-CoV-2 was discerned for its respiratory symptoms but as time goes by, research progressively reveals that COVID-19 can result in long-term health complications [8]. The persisting and perpetually lingering effects of COVID-19, particularly in individuals with persistent symptoms post-infection, have become a subject of unequivocal medical concern [9] with regard to a possible link between COVID-19 and lung cancer. In this review, we delve into the possible link between COVID-19 infection and lung cancer, exploring persistent symptoms, long-term respiratory sequelae, and possible molecular mechanisms that could predispose patients to lung malignancies.

## 2. Persistent Symptoms and Long-Term Respiratory Sequelae in COVID-19 Survivors

“Long COVID” refers to persistent symptoms of post-acute sequelae of SARS-CoV-2. These perpetual symptoms are identified amongst some patients after recovering from acute COVID-19. These symptoms may include cough, chest pain, fatigue, dyspnea, and “brain fog” or cognitive dysfunction [8,10]. Research studies have shown that approximately 10–30% of COVID-19 survivors experience long-term symptoms beyond three months post-infection [11]. Long COVID-19 symptoms may also consist of autonomic, cardiovascular, renal, and connective tissue dysfunctions [12]. Among the persistent symptoms, respiratory issues such as a chronic cough, reduced lung capacity, and interstitial lung changes are of significant concern [9,13]. The persistence of such symptoms indicates lasting damage to the lungs, therefore raising concerns around these individuals being potentially at increased risk of long-term respiratory conditions, including malignancies like lung cancer.

Many COVID-19 survivors exhibit long-term respiratory complications. These complications persist even after the acute phase of the illness has passed. A growing body of research highlights the development of fibrotic lung changes, particularly in those who experienced severe COVID-19 pneumonia or required mechanical ventilation [14]. Imaging studies have shown that ground-glass opacities (GGOs), pulmonary fibrosis, and persistent interstitial abnormalities can remain for months after recovery from various pulmonary diseases, including COVID-19 and idiopathic pulmonary fibrosis [15,16]. These abnormalities are often detected through high-resolution CT scans and are closely associated with the progression of interstitial lung diseases. GGOs appear as hazy areas on imaging and are common in conditions like pneumonia or early-stage fibrosis [17].

In patients recovering from severe respiratory conditions, these opacities may persist, often accompanied by fibrotic changes such as honeycombing and traction bronchiectasis. Pulmonary fibrosis, which involves the thickening and scarring of lung tissue, can progress over time, leading to functional decline even after recovering from the main phase of illness [15]. Studies indicated that persistent interstitial abnormalities, like subpleural reticulations and traction bronchiectasis, increase the risk of long-term fibrosis. In some cases, these findings can indicate progression towards more severe lung diseases, even in asymptomatic individuals, making radiological follow-up crucial for the early detection of potential complications [16]. While the exact prognosis of these changes remains uncertain, the fibrotic remodeling of the lung tissue adds more vulnerability to trigger lung tissue carcinogenesis. Long-lasting inflammation and cellular damage may incidentally enhance cell malignancy [18,19]. These findings underscore the potential for COVID-19 to leave lasting effects on lung tissue, elevating the risk of tumorigenesis.

A study conducted in 2023 reveals that patients who recovered from severe COVID-19, especially those admitted to the intensive care unit (ICU), are more at risk of being diagnosed with cancer. Indeed, the study compares the incidence of cancer in two groups: 41,302 individuals admitted to the ICU due to SARS-CoV-2 and 713,670 control individuals not hospitalized for SARS-CoV-2. During the follow-up, it was found that 2.2% of the ICU-group patients were diagnosed with cancer in the following months, compared to 1.5% in the control group. The study reveals a significantly high risk of lung cancer and other types of cancers such as renal cancer, hematological cancer, and colon cancer among the ICU-group patients. The study suggests that severe COVID-19 infection could be considered as an indicator of undiagnosed cancer [20]. Another less recent study assessed the lung CT scan of COVID-19 survivors after 3 months and 6 months from the infection. Researchers found that lung cancer was most common after three months from the infection, and identified in patients who had severe COVID-19 infection with severe ARDS and massive fibrosis with smoking history. The study also showcased that the most common type of lung cancer was squamous-cell lung cancer [21].

## 3. Lung Cancer

Lung cancer unequivocally is the principal cause of cancer-related deaths worldwide and is characterized by the uncontrolled and thereby uninhibited growth of abnormal cells in the lungs [22]. There are mainly two types of lung cancer: non-small-cell lung cancer (NSCLC), which represents about 85% of the cases, and small-cell lung cancer (SCLC), which represents about 15% of the cases and is considered more aggressive but less common [23]. In total, 40% of NSCLCs are adenocarcinomas, 25 to 30% are squamous-cell carcinomas, and the remaining 10 to 15% are large-cell carcinomas [24].

Lung cancer risk factors primarily include smoking, exposure to carcinogens such as asbestos, and a history of lung diseases such as tuberculosis, pneumonia, asthma, chronic bronchitis, and chronic obstructive pulmonary disease (COPD) [25]. Symptoms related to lung cancer often appear late during the progression of the disease and may include a persistent cough, chest pain, hemoptysis, and unexplained weight loss. Early diagnosis and detection are crucial to improve prognosis, despite the fact that the vast majority of cases are somehow diagnosed at an advanced stage [26].

## 4. Is There a Link Between COVID-19 Infection and Lung Cancer?

Several studies have demonstrated that patients with pre-existing lung conditions such as COPD or pulmonary fibrosis, both associated with lung cancer, are more vulnerable to severe outcomes of COVID-19, potentially amplifying the risk of cancer [18,27]. Another study supposed that SARS-CoV-2 has a direct impact on the growth and outcome of cancer after finding the RNA of this virus in lung cancer cells metastatic to the brain [28].

Nevertheless, a direct link between COVID-19 infection and lung cancer is yet to be ascertained, and such scientific development remains elusive and under investigation. There is a growing urge within the research community to explore the long-term impact of COVID-19 on lung tissues whereby prompting investigations into whether it could contribute to the risk of lung cancer. SARS-CoV-2 triggers significant lung inflammation through cytokine storms, causing tissue damage, fibrosis, and immune dysregulation.

### 4.1. Cytokine Storm and Chronic Inflammation

The molecular mechanism which links COVID-19 infection to lung cancer remains speculative. However, studies center around the inflammatory and immune pathways which are activated during the infection. Chronic inflammation is considered to be a certain risk factor for cancer, wherein prolonged inflammation in the lungs may facilitate tumorigenesis [29]. An intense immune reaction which is known as a “cytokine storm” occurs during the severe COVID-19 infection phase [30,31]. Cytokine storm is a life-threatening systemic inflammatory syndrome implicating an increase in circulating cytokines and immune-cell hyperactivation that can be generated by pathogens, cancers, or autoimmune conditions. Although some cytokines are necessary in controlling the infection, they are harmful to the host [32]. Consequently, a cytokine storm induces a cytokine cascade, which includes interferon-gamma (IFN-γ), tumor necrosis factor α (TNF-α), interleukin-1 (IL-1), interleukin-6 (IL-6), and interleukin-18 (IL-18), all of which have significant roles in the toll-like receptor signaling pathway [33].

IL-6, TNF-α, and IL-1β cytokines become exceedingly and excessively elevated, eventually leading to prolonged inflammation during the infection and recuperation phase [34]. Severe and chronic inflammation can induce oxidative stress and subsequent DNA damage, which unequivocally have an important role in the development of cancer [35].

The incremental increase in cytokines may eventually cause ubiquitous tissue damage, such as acute lung injury [36]. IL-6 cytokine, in particular, is known to activate the Janus kinase (JAK)/signal transducer and activator of transcription 3 (STAT3) pathway, which eventually amplifies cell proliferation and survival, processes associated with tumorigenesis [37]. It was also shown that IL-6 was abnormally hyperactivated in different cancers [38]. Obviously, IL6 elevation in both cancer and COVID-19 does not necessarily mean that there is a link between them. However, there seems to exist a possibility of interplay of COVID-19 and cancer in IL-6/JAK/STAT signaling. IL-6 is responsible for inducing and triggering the release of a large number of pro-inflammatory cytokines in the tumor. This chronic inflammatory environment then contributes to the development of cancer [39]. Researchers in the scientific community have documented that IL-6 plays a crucial role as a driver of tumor progression and can also act as a biomarker for diagnosing and predicting the prognosis of cancer [38,40]. Elevated levels of IL-6 have been observed in patients affected by different types of cancer, including pancreatic, breast, prostate, colorectal, and NSCLC [41,42,43,44]. On the other hand, JAK1 is responsible for STAT3 activation in lung cancer cells. It was shown that using IL-6 neutralizing antibody can inhibit lung cancer growth [45].

The classic pathway of IL-6/JAK/STAT signaling starts with the binding of IL-6 to its receptor IL-6 receptor (IL-6R) on the cell membrane and successive interaction with the transmembrane protein IL-6 receptor subunit-beta (gp130, named also IL-6Rβ).

The activation of the JAK enzyme is realized due to gp130 action. Later, the JAK enzyme will phosphorylate tyrosine residues on gp130, which will offer a docking site for the STAT3 protein, initiating downstream signaling. When the phosphorylated gp130 and the STAT3 protein bind, the JAK enzyme will phosphorylate it, leading to its homo-dimerization. Afterwards, the new dimer moves to the nucleus, where it eventually enhances the transcription of several genes [46].

There is also another pathway called the trans-signaling pathway which consists of the binding of IL-6 to a soluble form of IL-6R (sIL-6R), leading to the formation of a complex between IL-6-sIL-6R and gp300 [47].

It is worth noting that IL-6-induced neoplastic changes can also result from other factors rather than COVID-19 infection such as cigarette smoking and exposure to carcinogens. These factors could provoke mutations in the Kirsten Rat Sarcoma Virus gene (KRAS), which in turn increases the expression levels of IL-6 in the lung, resulting in adenocarcinoma pathogenesis via the JAK/STAT3 pathway [48,49].

### 4.2. SARS-CoV-2 Could Be an Oncogenic Virus

The increased frequency and thereby incidence of lung cancer amongst COVID-19 patients is likely ascribed to the severe suppression of the immune system caused by SARS-CoV-2 [20]. Thereby the changes are related to the inflammatory components and the cascade of immunogenic events activated by SARS-CoV-2, and, finally, there exists a possibility that the virus is oncogenic.

Indeed, approximately 12% of all human cancers, in all probability, are potentially the result of viral infections [50]. One of the hypotheses regarding the oncogenic nature of SARS-CoV-2 is hijacking the function of the p53 protein. According to this hypothesis, SARS-CoV-2 would have developed strategies similar to Epstein–Barr virus (EBV) and hepatitis B virus (HSV1) for eventually controlling the p53 protein by hijacking its function via virus antigens, leading to its degeneration [51]. The data that could consolidate this hypothesis were extracted from the mechanisms of SARS-CoV-1 which has around 89% homology with SARS-CoV-2 [51]. One of the mechanisms of p53 function hijacking is performed by the non-structural protein2 (Nsp2) of SARS viruses. NSP2 interacts with prohibitin 1 and 2 (PHB1 and PHB2), which are primarily localized in the mitochondrion and play a central role in continuing mitochondrial DNA activity [51]. PHB1 and PHB2 cooperate together and provide a scaffold for the spatial organization of enzymatic activities in mitochondria. The depletion of PHB1 and PHB2 in the cell can lead to a series of responses, ultimately causing the liberation of reactive oxygen species (ROS) into the cell nucleus, resulting in oxidative stress. This process can also hinder the activation of p53-dependent genes, contributing to cellular impairment [52]. In addition, the two proteins have an important role in the nucleus. They can bind as cofactors with various epigenetic regulators and transcription factors, including p53, for which they act both as an activator and a chaperone. Thus, prohibitin reduction impairs the transcriptional activities of p53 [52].

Another mechanism of TP53 hijacking could be realized by the non-structural protein 3 (Nsp3) viral protein. Indeed, the papain-like protease of NSP3 interacts with the E3 ubiquitin ligase ring-finger and CHY zinc-finger domain-containing 1 (RCHY1), thus increasing the RCHY1-mediated degradation of p53. The weakening of p53 can be seen as an adaptive strategy of SARS-CoV-2, allowing it to exploit the cell pathways controlled by p53 for its own replication during the acute phase of infection. This manipulation enables the virus to evade the host’s immune response and promotes its replication [53].

Another hypothesis regarding the mechanism through which SARS-CoV-2 may have a potential oncogenic effect is acting as human papilloma virus (HPV). In this scenario, the non-structural protein 15 (NSP15) and NSP3 proteins would inhibit tumor suppressors such as P53 and retinoblastoma protein (PRB). This inhibition could be one of the factors leading to cancer. NSP15 and NSP3 in SARS-CoV-2 would play the same role as E6 and E7 in HPV, binding and inhibiting the two tumor suppressors P53 and PRB [54]. The available data consolidating this hypothesis are the inhibition of TP53 by NSP3 and the interaction of NSP15 with PRB in SARS-CoV-1 [53,55]. The interaction of NSP15 with PRB induces nuclear export and ubiquitination, targeting PRB for proteasomal degradation [55].

Both hypotheses should be verified experimentally using animal models. Nevertheless, whatever the hypothesis, there is one constant fact, which is the high and persistent reduction in TP53 after severe SARS-CoV-2 infection. Indeed, a previous study demonstrated that, in severely affected and critical patients with long COVID, the majority of the pathways associated with TP53, such as apoptosis, DNA damage response, and signal transduction, were remarkably down-regulated compared to non-severe and healthy control groups [56]. As expected, the down-regulation of TP53 significantly impacted a number of interacting genes.

### 4.3. COVID-19 Infection Is a Cause of DNA Damage and Impaired DNA Repair

COVID-19 infection can potentially lead to DNA damage and genome instability during its replication process in mammalian cells. This can result in the deregulation of the cell cycle [57]. The initiation of DNA damage and aberrant DNA repair mechanisms is closely linked to the development of chronic diseases such as diabetes, neurodegenerative disorders, and cancer [58].

As mentioned above, SARS-CoV-2 infection will induce the activation of pro-inflammatory cytokines such as IL-6, TNF-α, and interferons. The prolonged inflammation, which is caused by monotonously repeated immune activation, can eventually lead to DNA damage, disrupted and hampered cellular repair mechanisms, and finally, increased cellular proliferation; all of the above are major key factors in tumorigenesis. The cytokine storm can literally cause excessive and thereby extensive tissue damage and fibrosis, which, in turn, creates a microenvironment conducive to the development of cancer. Unequivocally, viral infections can induce tumorigenesis through DNA damage either by chronic inflammation or by mechanisms related to viral replication processes that can contribute to genotoxicity [58].

Several arguments and clues support the potential predisposition of COVID-19 survivors to lung cancer. This predisposition may happen through several molecular mechanisms, which duly involve damaging the DNA, thereby creating genomic instability and, eventually, causing the deregulation of the cell cycle. In the same context, a 2023 study demonstrated that DNA damage was realized via Checkpoint kinase 1 (CHK1) degradation and impaired p53-binding protein 1 (53BP1) recruitment. Indeed, according to the study, Open Reading Frame 6 (ORF6) and nonstructural protein 13 (NSP13) viral proteins will be able to impair CHK1 DNA damage response through proteasome and autophagy. The lack of CHK1 will induce dNTP shortage, which will naturally cause reduced S-phase progression, DNA damage, the activation of pro-inflammatory pathways, and cellular senescence [59].

DNA damage can also result from NS13 interaction with DNA polymerase δ, leading to DNA replication fork stress [60]. The defects in the replication fork lead to H2AX histone phosphorylation and cell cycle arrest, which can promote tumorigenesis by inducing genetic instability [61]. Genetic instability includes genetic alterations such as base changes, insertions and deletions, and chromosomic rearrangements. Usually, the errors that could happen during DNA replication can be repaired by DNA polymerase δ. This enzyme also plays a role in synthesizing the lagging strand during replication. The interaction of NS13 with DNA polymerase δ will impair its functions, leading to DNA damage [62].

Based on all the above-mentioned factors, it can be speculated and thereby predicted that SARS-CoV-2 may directly interact with oncogenes or tumor suppressor genes; however, supplementary investigation is needed to duly understand and eventually elucidate these pathways.

Additionally, the generation of ROS during the infection and inflammation phase further contributes to DNA damage, thereby increasing the likelihood of oncogenic mutations [35].

SARS-CoV-2 infection induces cellular stress by disrupting normal cellular functions. Consequently, this will lead to DNA damage and simultaneously impair the DNA repair mechanisms. Research studies have shown that viral infections, including COVID-19, may disrupt cell cycle checkpoints and the activity of proteins such as p53, which is responsible for maintaining genomic stability and integrity [58]. Due to the disruption of cell cycle checkpoints and the activity of p53 proteins, DNA mutations and genomic instability take place, which are key hallmarks of cancer development [63].

The SARS-CoV-2 virus may interfere with the DNA repair mechanisms and pathways, particularly affecting the homologous recombination (HR) and non-homologous end joining (NHEJ) pathways, which are crucial and thereby critical for the fixation of double-strand breaks in DNA [57]. The dysregulation in these pathways eventually leads to the accumulation of DNA errors and chromosomal aberrations, thereby stimulating and fostering a conducive and favorable environment suitable for the initiation of cancer [64].

The impairment of DNA repair can happen through two mechanisms. First, the depletion of CHK1 leads to the loss of the ribonucleoside-diphosphate reductase subunit (RRM2), which results in a reduction in cellular levels of dNTPs and DNA replication stress. Second, the SARS-CoV-2 nucleocapsid (N) protein binds to damage-induced long non-coding RNAs (dilncRNAs), which results in the inactivation of 53BP1 and defects in DNA repair [65].

Another indication that SARS-CoV-2 impairs the DNA repair mechanisms is the complete inhibition of the replication of SARS viruses in epithelial cell lines by the ATR inhibitor (Berzosertib). This is a supplementary evidence of the dependance of SARS-CoV-2 replication to ATR-dependent DNA repair mechanisms [66].

### 4.4. Endoplasmic Reticulum (ER) Stress and Unfolded Protein Response (UPR)

Previous studies showed that SARS-CoV was able to induce ER stress [67]. Furthermore, SARS-CoV-2 infection can initiate and cause ER stress courtesy of the high requirement and thereby demand for the production and folding of proteins during the replication of the virus [68]. Indeed, the virus partially activates the inositol-requiring enzyme1α (IRE1α)/X-box binding protein 1 (XBP1) ER stress pathway in human lung cells [68].

In a normal situation, UPR can either succeed in refolding correctly the accumulated unfolded proteins or fail to do so, which will lead to its degradation by the ubiquitin-proteasome pathway. When the unfolded proteins exceed the threshold, apoptosis will be activated. Apoptosis is mediated by activating transcription factor 4 (ATF4) and activating transcription factor 6 (ATF6), and the activation of the JNK/AP-1/Gadd153-signaling pathway [69]. The prolonged and sustained dysregulation of UPR can eventually lead to apoptotic resistance, which is a feature of cancer cells [70]. ER stress induces and thereby promotes inflammation, which activates pathways related to oncogenesis such as NF-κB, which eventually contributes to the development of tumors [71].

In sum, the ability of COVID-19 to initiate and thereby induce chronic inflammation, damage to DNA, genomic instability, the deregulation of mechanisms involved in DNA repair, and UPR creates a favorable and thereby conducive environment for the development of lung cancer in patients who have recovered (Figure 1). These effects underscore the need for a long-term follow-up of post-COVID-19 patients, particularly patients with some form of lung abnormality.

### 4.5. COVID-19 and ACE2 Receptor

SARS-CoV-2 utilizes the angiotensin-converting enzyme 2 (ACE2) receptor to enter lung cells. Indeed, the virus binds to the ACE2 receptor using the spike protein which is localized at its capsule. ACE2 is the primary receptor for SARS-CoV-2, which is highly expressed in the epithelial cells of the lung [72]. As ACE2 usually protects against lung injury by the regulation of the renin–angiotensin system, the virus hijacks this receptor, thereby disrupting lung homeostasis. The down-regulation of ACE2 following infection may impair the repair and regeneration of lung tissue, thereby creating an environment which is susceptible and prone to fibrotic changes and chronic cellular damage, which eventually increases the risk of cancer [73]. The hypothesized scenario regarding the potential role of SARS-CoV-2 using the ACE receptor is the reduction in free ACE receptor. This reduction will increase the accumulation of AngII, which would aggravate the inflammatory response. In natural conditions, ACE2 degrades AngII, in alveolar epithelial cells, in Ang (1–7). Therefore, the accumulated Angiotensin II (AngII) would bind to NF-κB to stimulate the expression of inflammatory cytokines. The reduced levels of Ang (1–7) would lead to the activation of PI3K/Akt and ERK signaling pathways and participation in tumor development [74]. This hypothesis needs to be checked by experimental studies.

## 5. SLC22A18, ZEB2, and ZEB1 as Molecular Markers of Lung Cancer That Could Be Overexpressed by COVID-19 Infection

Solute carrier family 22 member 18 (SLC22A18) is a transporter for cationic organic solutes such as chloroquine and quinidine [75]. One study showed that the SLC22A18 gene is overexpressed in lung cancer, and its expression is correlated with pathological grade [76]. A previous study, on a large cohort of NSCLC patients, showed that miR-137 was drastically overexpressed, which suppressed proliferation and migration in NSCLC patients [77]. Similarly, when miR-137 expression was reduced, it promoted SLC22A18 expression and tumor aggressivity [78]. In the other hand, an imprinting loss of SLC22A18 leads to its overexpression in NSCLC tissues [79]. Based on miRNA-137 interactions, we can conclude that SLC22A18 has a potential oncogenic role and may be useful as a diagnostic and prognostic biomarker in NSCLC [78].

The expression of SLC22A18 can be stimulated by zinc finger E-box-binding homeobox 2 (ZEB2) transcription factor, which is, in turn, stimulated by zinc finger E-box-binding homeobox 1 (ZEB1), another transcription factor that was found to be overexpressed in lungs after COVID-19 infection [80]. Additionally, SARS-CoV-2 infection increases the expression of ZEB1 associated with EMT, further highlighting the impact of the virus on cancer-related processes [81].

The transcription factor ZEB1 has an important role in altering the expression of epithelial genes in lung cancer, including SEMA3F. ZEB1 is closely linked to the mesenchymal phenotype of NSCLC. Furthermore, ZEB1 is essential for the development of the pulmonary mesenchymal cancer phenotype [82]. It was also found that the epithelial–mesenchymal transition is associated with an elevated expression of ZEB1, which correlates with different grades and stages of lung cancer [83].

Further functional analysis could be used to investigate the pathway that involves ZEB1, ZEB2, and SLC22A18. We suggest using a QRT-PCR to detect the expression of ZEB1, ZEB2, and SCL22A18 in NSCLC tissue samples. Kaplan–Meier analysis could also help to determine if there is an association between the genes’ expression and the survival rate of NSCLC patients. In addition, experiments using an animal model should be conducted in order to measure the expression levels of ZEB1, ZEB2, and SLC22A18 after COVID-19 infection and to determine if their increase will induce tumorigenesis. Moreover, using cell lines, the measurement of the expression levels of ZEB1, ZEB2, and SLC22A18 in NSCLC cell lines and an MTT assay could determine their influence on cell proliferation [84].

## 6. Challenges in Proving Causality Between COVID-19 and Lung Cancer

This review reports clinical and epidemiological studies showing that lung cancer was diagnosed in COVID-19 survivors who had severe infection [20]. This, therefore, points to a possible relationship between the two pathologies. Clinical data also show that fibrotic lung changes, which are long-term COVID-19 symptoms, may add more vulnerability to cause lung tissue carcinogenesis [14]. These solid data incite us to study the molecular pathways that are shared between COVID-19 infection and lung cancer. In other words, the question would be how this infection can trigger lung cancer. Although many pathways were scientifically proven, as is the case with the JAK/STAT3 pathway and its induction through a cytokine storm and mainly through IL-6 hyperproduction, it is believed that other external factors like smoking and polluted air will also play a contributive role in initiating the cancer [37,85]. Despite the fact that TP53 was reduced in COVID-19 survivors for a long period, the oncogenic nature of SARS-CoV-2 remains hypothetic and further experimental studies need to be realized to prove that [51]. Undoubtedly, DNA damage, impaired DNA repair mechanisms, ER stress, and UPR are consequences of the infection by the novel coronavirus [58,68]. However, this does not mean that the viral infection necessarily degenerates into cancer. Finally, ACE receptors are not found only in lungs. They can be found in intestinal epithelial cells of the gut and in endothelial and smooth cells of the blood vessels, heart, and kidneys [86]. In other words, cancer can potentially occur in these organs. However, the focus of this review are the lungs, as they have high trophism for SARS-CoV-2 and are the first suspected site for cancer to potentially develop. Further experimental studies need to be realized to confirm the molecular pathways that can lead to lung cancer after a severe COVID-19 infection and the probable external and internal contributors or risk factors.

## 7. Conclusions

Severe COVID-19 infection may serve as an early marker for lung cancer development, particularly pronounced in individuals with risk factors and predisposing conditions thereof. The post-infection inflammatory response and immune dysregulation triggered and thereby initiated by the virus may exacerbate pre-existing lung damage or may act as a catalytic factor for the eventual initiation of carcinogenesis in vulnerable individuals. Furthermore, genetic mutations caused by the hypoxia and oxidative stress associated with severe respiratory failure during COVID-19 can further increase the risk and thereby chances of malignant transformation. It is thus pertinent that patients who eventually recover from severe COVID-19 infection should undergo mandatory screening and continual monitoring for potential malignancies, particularly if long-term respiratory sequelae are present.

Based on the above-mentioned data, severe COVID-19 infection, smoking, exposure to polluted air, and previous respiratory conditions such as COPD are the main risk factors for lung cancer.

As the globe still struggles with the long-term repercussions of the COVID-19 pandemic, understanding myths and realities associated with COVID-19 and its probable role as one of the potential risk factors for lung cancer is all the more critical. A definitive link established between COVID-19 and lung cancer is still elusive. The evidence of long-term respiratory sequelae, perpetual and persistent inflammation, and, finally, the molecular pathways initiated and triggered by SARS-CoV-2 suggest a probable association and a potential link. The prolonged monitoring of recovered COVID-19 survivors, who were plagued with severe infections or pre-existing lung conditions, for signs and symptoms of malignancy is advisable. Focused and continued research activity is essential to clarify underlying mechanisms and develop appropriate screening mechanisms. Innovative strategies using AI need to be conceptualized and implemented for the prevention of illness in populations who are at risk.

## Figures and Tables

**Figure 1 pathogens-13-01070-f001:**
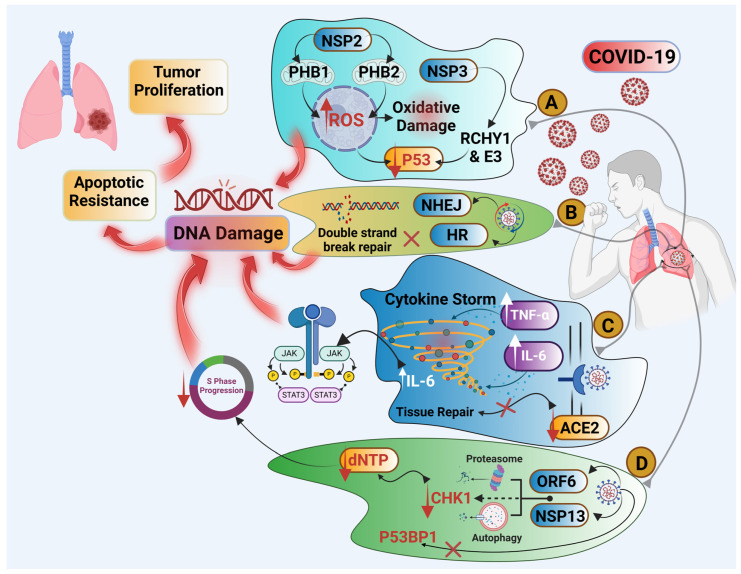
Probable lung cancer pathways induced by COVID-19 infection. (**A**) The pathway of SARS-CoV-2 as oncogenic virus, hijacking P53 and leading to oxidative stress and DNA damage. (**B**) SARS-CoV-2 interfering with DNA repair mechanisms and causing DNA damage. (**C**) The JAK/SAT3 pathway is activated by a cytokine storm due to IL-6 increase; similarly, we see the down-regulation of ACE2 receptor, leading to the impairment of tissue repair. (**D**) S-phase deregulation through ORF6 and NSP13 interactions leading to the disruption of cell cycle and DNA damage. All these events may lead to tumorigenesis. Up arrows: increase and down arrows: decrease.

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
