# Peer review of "Equivocating and Deliberating on the Probability of COVID-19 Infection Serving as a Risk Factor for Lung Cancer and Common Molecular Pathways Serving as a Link"

_pathogens, 2024, doi:10.3390/pathogens13121070_

Round 1
Reviewer 1 Report
Comments and Suggestions for Authors
This review considers evidence for a link between COVID-19 and lung cancer. This is a significant possibility and is worthy of consideration in a review for Pathogens. However, this manuscript suffers from limitations in the authors' knowledge of English. In particular the language is very florid (for example lines 324-6) and some is meaningless. For example the phrase 'in all earnest' is used several times and makes no sense. Basically, the manuscript must be corrected/modified by an English speaker.
A number of specific changes should also be made:
Lines 105-9 is repetitive.
Lines 204-211 is rather confused and needs to be clarified.
Because IL6 is elevated in cancer and SARS infection it does not mean that there is a link between the two. The discussion of IL6 should make this clear.
Sections 4.3 and 4.4 should be amalgamated.
Section 5 is highly speculative and could be omitted.
Author Response
Comment 1: This manuscript suffers from limitations in the authors' knowledge of English. In particular the language is very florid (for example lines 324-6) and some is meaningless. For example, the phrase 'in all earnest' is used several times and makes no sense. Basically, the manuscript must be corrected/modified by an English speaker.
Answer: ‘’In all earnest’’ was removed from all sentences.
432-434: the paragraph is now less ornamental.
The manuscript was corrected by a native speaker.
Comment 2: A number of specific changes should also be made: Lines 105-9 is repetitive
Answer: The Lines 105-9 have been removed
Comment 3: Lines 204-211 is rather confused and needs to be clarified.
Answer: These lines were modified to be clearer (lines 222-228).
Comment 4: Because IL6 is elevated in cancer and SARS infection it does not mean that there is a link between the two. The discussion of IL6 should make this clear.
Answer: We appreciate the reviewer’s comment. We have added one sentence clarifying that ‘’IL6 elevation in both cancer and COVID-19 does not necessarily mean that there is link. However, there seems to exist a possibility of the interplay of COVID-19 and cancer in IL-6/JAK/STAT signaling’’ (lines 168 -169). Moreover, we explained how this can happen in molecular terms (lines 180-196).
Comment 5: Sections 4.3 and 4.4 should be amalgamated.
Answer: Sections 4.3 and 4.4 were merged together and new information were added (lines 247 to 310).
Comment 6: Section 5 is highly speculative and could be omitted.
Answer: We appreciate the suggestion of the reviewer. However, we propose to keep this part because it presents a new and interesting hypothesis. Although, it is speculative, we tried to add more data to support it and suggest some experiments to verify it in future studies (lines 393 to 415).
Reviewer 2 Report
Comments and Suggestions for Authors
Dear Author, I give you the following comment. Please address this in your manuscript to enhance the readability and understanding of your manuscript.
Major Comment Questions:
- How do the authors justify the potential link between SARS-CoV-2 infection and the development of lung cancer, considering the absence of long-term clinical data on the subject?
- The article mentions the JAK/STAT3 pathway as a common link between SARS-CoV-2 infection and lung cancer. Could the authors elaborate on how these pathways specifically interact in lung tissue during COVID-19 infection?
- The potential role of the p53 protein in SARS-CoV-2-induced oxidative stress and its relation to DNA repair mechanisms needs further clarification. Can the authors provide more mechanistic details on how p53 might be hijacked by the virus?
- Could the authors discuss potential confounding factors that might influence the development of lung cancer in COVID-19 patients, such as pre-existing conditions or environmental exposures?
- The overexpression of SLC22A18 and ZEB1 during COVID-19 infection is an interesting hypothesis. How do the authors propose to experimentally verify these findings in future studies?
Minor Comment Questions:
- The abstract briefly mentions the JAK/STAT3 pathway without much detail. Could the authors clarify the specific cellular events triggered by this pathway in lung cells during SARS-CoV-2 infection?
- The article refers to "genomic instability" in the context of SARS-CoV-2 infection. Could the authors expand on what types of genomic instability might occur and how they relate to lung cancer development?
- The statement about p53 hijacking is intriguing, but how does this process compare to other viruses known to interfere with p53, such as HPV?
- The concept of cytokine storm is central to the paper. Could the authors explain in more detail how the cytokine storm directly contributes to lung cancer risk?
- The article suggests a link between COVID-19 and lung cancer through molecular mechanisms. Can the authors provide a brief discussion of the challenges in proving causality in this potential relationship?
These questions aim to address both overarching concerns and specific technical details that could impact the robustness and clarity of the study's findings.
Best Regards
Comments on the Quality of English LanguageFine
Author Response
Major Comment Questions:
Comment 1: How do the authors justify the potential link between SARS-CoV-2 infection and the development of lung cancer, considering the absence of long-term clinical data on the subject?
Answer: We appreciate the comment of the reviewer and we want to clarify that we have already mentioned that there is one clinical study which demonstrated that patients who had severe COVID-19 are more likely to be diagnosed with lung cancer or other types of cancer (lines 102-111). Nevertheless, to consolidate more the point, we added another study that confirmed that survivors of severe COVID-19 presented with a diagnosis of lung cancer after three and six months of the infection (lines 111-6).
Comment 2: The article mentions the JAK/STAT3 pathway as a common link between SARS-CoV-2 infection and lung cancer. Could the authors elaborate on how these pathways specifically interact in lung tissue during COVID-19 infection?
Answer: We thank the reviewer for the valuable comment. Based on his request, we elaborated more how the JAK/STAT3 could be a common link between COVID-19 infection and lung cancer (lines 180-196). The relationship between IL-6 and JAK/STAT3 pathway was also elaborated (lines 192-196).
Comment 3: The potential role of the p53 protein in SARS-CoV-2-induced oxidative stress and its relation to DNA repair mechanisms needs further clarification. Can the authors provide more mechanistic details on how p53 might be hijacked by the virus?
Answer: We thank the reviewer for this comment. Based on his request, we added further clarifications about the mechanistic details of hijacking the p53 protein by the SARS-CoV-2 virus and inducing the oxidative stress which could impair DNA repair mechanisms (lines 210-221).
Comment 4: Could the authors discuss potential confounding factors that might influence the development of lung cancer in COVID-19 patients, such as pre-existing conditions or environmental exposures?
Answer: It is true that other factors such as smoking, preexisting respiratory diseases and exposing to pollution are the main risk factors to lung cancer. These factors have been already cited in the lines 127-129. We have added after the revision two new lines (from 399 to 403) to highlight that the severe COVID-19 infection associated to the other cited factors would increase the risk of lung cancer development.
Comment 5: The overexpression of SLC22A18 and ZEB1 during COVID-19 infection is an interesting hypothesis. How do the authors propose to experimentally verify these findings in future studies?
Answer: We strongly appreciate the comment of the reviewer. We have added few points to strengthen the hypothesis and suggested, as requested by the reviewer, some ideas to verify experimentally the hypothesis of overexpression of SLC22A18, ZEB2 and ZEB1 during COVID-19 infection (lines 354-361, 365,367 and 374-382).
Minor Comment Questions:
Comment 1: The abstract briefly mentions the JAK/STAT3 pathway without much detail. Could the authors clarify the specific cellular events triggered by this pathway in lung cells during SARS-CoV-2 infection?
Answer: A new sentence regarding, JAK/STAT3 pathway, was added in the abstract (lines 23-24)
Comment 2: The article refers to "genomic instability" in the context of SARS-CoV-2 infection. Could the authors expand on what types of genomic instability might occur and how they relate to lung cancer development?
Answer: Some new information was added to expand more the genomic instability that occurs due to SARS-CoV-2 infection and that can lead to lung cancer (lines 263-266 and 273-280).
Comment 3: The statement about p53 hijacking is intriguing, but how does this process compare to other viruses known to interfere with p53, such as HPV?
Answer: A clear explanation of the two mechanisms by which SARS-CoV-2 would hijack p53 function was added to the manuscript (line 203-228). In addition, we elaborated another hypothesis that pretends that SARS-CoV-2 and HPV may have similar mechanisms in interfering with tumor suppressors such as p53 and thus a similar oncogenic nature (line 229-238).
Comment 4: The concept of cytokine storm is central to the paper. Could the authors explain in more detail how the cytokine storm directly contributes to lung cancer risk?
Answer: Actually, the concept of cytokine storm was extensively detailed in the paper (line 161-165). Nevertheless, in order to clarify more the concept, we added a short definition of cytokine storm (line 151-155). We also added an explanation on how the cytokine storm, through elevation IL-6 levels, can contribute to the lung cancer risk (line 180-191).
Comment 5: The article suggests a link between COVID-19 and lung cancer through molecular mechanisms. Can the authors provide a brief discussion of the challenges in proving causality in this potential relationship?
Answer: We appreciate the reviewer’s suggestion. A new part entitled ‘’Challenges in proving causality between COVID-19 and lung cancer ‘’ has been added in the paper (line 393-415).
Round 2
Reviewer 1 Report
Comments and Suggestions for Authors
The English is still a little florid and inappropriate in a few cases but the manuscript is much improved and I would recommend publication in its present form.